# The Development of a Program to Identify and Manage Apathy in Residents with Korsakoff’s Syndrome: A Qualitative Exploration of Patient, Family Caregiver, and Professional Caregiver Perspectives

**DOI:** 10.3390/geriatrics10060146

**Published:** 2025-11-06

**Authors:** Maud E. G. van Dorst, Julia Roosenschoon, Johanna M. H. Nijsten, Annette O. A. Plouvier, Raymond T. C. M. Koopmans, Debby L. Gerritsen, Yvonne C. M. Rensen, Roy P. C. Kessels

**Affiliations:** 1Centre of Excellence for Korsakoff and Alcohol-Related Cognitive Disorders, Vincent van Gogh Institute for Psychiatry, 5803 DN Venray, The Netherlands; 2Department of Neuropsychology and Rehabilitation Psychology, Donders Institute for Brain, Cognition and Behaviour, Centre for Cognition, Radboud University, 6525 XZ Nijmegen, The Netherlands; 3Department of Neurosurgery, Radboud University Medical Center, 6525 GA Nijmegen, The Netherlands; 4Archipel, Landrijt Expertise Centre for Specialized Care, 5623 ME Eindhoven, The Netherlands; 5Department of Primary and Community Care, Research Institute for Medical Innovation, Radboud University Medical Center, 6525 EZ Nijmegen, The Netherlands; 6Radboudumc Alzheimer Center, Radboud University Medical Center, 6525 GA Nijmegen, The Netherlands; 7Tactus Addiction Care, 7418 ET Deventer, The Netherlands; 8Klimmendaal Rehabilitation Specialists, 6813 GG Arnhem, The Netherlands

**Keywords:** apathy, neuropsychiatric symptom, intervention, long-term care, Korsakoff’s syndrome

## Abstract

**Background**: Apathy is a neuropsychiatric symptom that is frequently present in nursing-home residents, including residents with Korsakoff’s syndrome (KS). Although apathy is common in KS, treatment guidelines are lacking. The Shared Action for Breaking through Apathy (SABA) program, developed for people with dementia, was previously shown to be feasible in that group. The applicability of this program for the KS population seems promising, yet it was expected that the program would need to be adapted. This study aims to 1) explore what is important in identifying and managing apathy in individuals with KS, and 2) investigate the appropriate adjustments to the SABA program. **Methods**: This qualitative study consisted of semi-structured interviews with people with KS (*n* = 3), family caregivers (one spouse and one sibling) and professional caregivers (two nurses), and a multidisciplinary focus group meeting with professional caregivers (*n* = 12) experienced in care for people with KS. The focus group meeting was performed to deepen the understanding of the interview findings and further explore recommendations for adjustments to the SABA program. Thematic analysis was used to process the data. **Results**: Addressing aim 1, two themes were identified: (1) the challenge to appraise signals of apathy, and (2) the challenge to assess the needs of people with KS. Based on these themes, specific adjustments were formulated to respond to aim 2. **Conclusions**: The themes that were identified in this study gave direction to a KS tailored SABA program, the feasibility of which needs to be studied next.

## 1. Introduction

Apathy is a neuropsychiatric symptom that can be described as a reduction in goal-directed activity in relation to behavior, cognition, emotions, and/or social interaction [1]. Apathy is a transdiagnostic symptom that is frequently present among different neurocognitive and neuropsychiatric disorders, including Korsakoff’s syndrome (KS). KS is a severe and irreversible neuropsychiatric disorder caused by thiamine depletion, often occurring in the context of chronic and excessive alcohol consumption [2,3]. KS is commonly known for its severe memory disorder and remarkable neurocognitive and neuropsychiatric symptoms. Besides apathy, confabulations and impaired awareness of and insight into deficits are also characteristic symptoms in KS, and moving to a long-term care facility is often inevitable [2,3,4].

Apathy impacts the lives of people with KS (PwKS) and their relatives. The presence of apathy in KS is associated with lower health-related quality of life, cognitive functioning, participation in daily activities, and social participation. Moreover, the presence of apathy is associated with the use of antipsychotic drugs [5]. Studies in people with dementia show that apathy also has a negative impact on health-related quality of life, well-being, the experienced burden of caregivers, and predicts mortality [6,7,8]. Despite this extensive evidence for the negative impact of apathy, approaches to treating apathy in KS are lacking.

There is broad consensus that apathy is a transdiagnostic symptom and that its clinical presentation is similar across different diseases or disorders [9]. A transdiagnostic view on apathy is beneficial for the improvement of fundamental research on the construct of apathy and for deepening the understanding of the underlying neuropathology. A cross-population literature review shows that—in addition to population-specific brain abnormalities underlying apathy—abnormalities within the frontostriatal circuitry, the anterior cingulate cortex, and the inferior parietal cortex are associated with apathy in different populations [10]. Furthermore, research on apathy across different patient populations may enhance the development and improvement of treatment methods to reduce apathy [11]. In a transdiagnostic literature review, we found great overlap in effective treatment methods that are used to reduce apathy in different populations [12]. In that literature review, we provided recommendations for the treatment of PwKS and apathy in long-term care facilities. It was suggested that interventions on apathy in KS must rely on external motivation and behavioral activation instead of intrinsic motivation and self-monitoring. Furthermore, interventions on apathy should be personalized and tailored to the participants’ capacities and needs. Finally, it was recommended to assess the nature and severity of the apathy in a specific individual before the intervention.

The Shared Action for Breaking Through Apathy (SABA) program [13] appears to show considerable overlap with the KS-specific recommendations identified in our review [14]. The SABA program is a theory- and practice-based intervention to empower family (FC) and professional (PC) caregivers. Aim of the intervention is to identify and manage apathy in people with dementia residing in long-term care facilities. The SABA program provides understandable information and enhances awareness of apathy, helps to overcome a knowledge gap, including inadequate expectations, and supports the skills of caregivers and family members to manage apathy. The rationale of the SABA program was based on qualitative research on the views on and experiences with identifying and managing apathy from the perspectives of people with dementia and their FCs and PCs [14]. The program consists of materials and procedures that are used throughout the different phases of the SABA program [13]. The SABA program was found to be applicable and feasible in dementia care in nursing homes. FCs and PCs felt empowered in maintaining meaningful contact with people with apathy [13].

In accordance with the recommendations formulated in our recent review [12], patient compliance with the SABA program does not rely on the intrinsic motivation of apathetic individuals themselves, as the program is directed at FCs and PCs who motivate and activate the residents when the apathy is present. Furthermore, the SABA program is designed as a customizable intervention that is shaped by personal features and preferences of the individuals with apathy. Finally, the recognition and assessment of apathy is a prominent part of the SABA program. We therefore expect that the SABA program, with specific adjustments, can also be applied in the care of PwKS who display apathy.

The current study expands on the findings of this apathy intervention in dementia [13,14], aiming to 1) further explore which themes and subthemes are important in identifying and managing apathy in PwKS from an integrated perspective (i.e., perspectives of PwKS, FCs and PCs from different settings), and 2) examine appropriate adjustments to the SABA program to create a KS-specific SABA intervention: the SABA-KS program.

## 2. Materials and Methods

### 2.1. Study Design

This explorative qualitative study was performed using individual semi-structured interviews with PwKS, their primary nurses (PC; professional caregivers) and loved ones involved in their life in a non-caregiver capacity (FC; Family caregiver), and a focus group meeting with PCs from various disciplines in the field of KS. This article will continue the use of the terms PC and FC introduced by Nijsten et al. [14]. The researchers who performed the interviews and led the focus group meeting were not involved in the care for these individuals with KS and did not know the participating PwKS. After exploring views on identifying and managing apathy in PwKS, in the interviews, the original SABA program was presented to the participating PwKS, their FCs and PCs, who were asked to suggest modifications to enhance the program’s suitability and usability for the KS population. In the focus group meeting, we further explored and contextualized the findings from the interviews and gathered important recommendations to adjust the SABA program and for the development of SABA-KS. The original SABA program was also presented during the focus group meeting. This study was reported following the Standards for Reporting Qualitative Research (SRQR) [15].

### 2.2. Semi-Structured Interviews

PwKS and apathy were recruited from two clinical institutes specialized in KS care, located in the southern part of the Netherlands: one being a diagnostic mental health care clinic and long-term care unit, the other a center of expertise of a long-term care organization. Primary practitioners of the different units were asked to select potential participants. One inclusion criterium was a diagnosis of alcoholic KS. All patients met the criteria for Alcohol-induced Major Neurocognitive Disorder, Amnesic-confabulatory Type outlined in the DSM-5 (291.1; i.e., Korsakoff’s syndrome [16]) and the criteria for alcoholic KS [17]. All patients had a history of thiamine deficit or malnutrition (with suspicions of a Wernicke encephalopathy), and a disproportionate amnestic disorder in comparison to all other cognitive functions. The cognitive deficits were verified by a neuropsychological assessment and were not due to another medical condition, (current) psychiatric disorder, or another substance use disorder. Participants all had a verified history of long-lasting alcohol misuse. Also, patients had to show signs of apathy observed by an FC and/or PC. All patients were at least 6 weeks abstinent from alcohol. Participants were recruited using purposive sampling based on sex, setting (mental health care and long-term care) and educational level. After including an individual with KS and apathy, one FC and one PC of that patient were also asked to participate. Formal caregivers were a nurse practitioner and a nursing assistant.

The interviews with all PwKS and one PC were conducted face-to-face. The other PC, the Patient Representative (PR) and both FCs were interviewed over video call. For the video calls the participants were asked to find a quiet and private area in their respective homes or workplaces. The interviewers conducted the calls from their office and had a neutral background. In all interviews, the researchers explicitly created a setting in which participants were stimulated to share all their relevant experiences and thoughts. Notes of any relevant observation were taken during each interview.

Two interviews (with a PwKS and their patient representative) were led by M.D., a trained interviewer and psychologist with seven years of experience working with PwKS, and supported by J.R., a trained qualitative researcher with no prior experience with PwKS. The other six interviews were conducted by J.R. alone to limit interrater variability. M.D.’s clinical knowledge and understanding of PwKS were combined with J.R.’s research perspective, creating discussions leading to a balance between interpretations and deductions, and objective findings from the conversational data. Neither M.D. nor J.R. had a therapeutic relationship with the participants.

The topic list was based on the topic list from the study of Nijsten et al. [14]. The first three audio recordings were reviewed by M.D. and J.R. to evaluate the topic list. After these three interviews, the topic list was slightly adjusted to include issues that emerged as important (see Appendix A for the definitive topic list).

### 2.3. Focus Group

Participants were recruited via the two institutions, and via the Korsakoff Knowledge Center (Korsakov Kenniscentrum; www.korsakovkenniscentrum.nl), a nationwide network of Korsakoff nursing-home facilities in the Netherlands. Calls for participation were made within the various professional groups, which yielded a limited number of responses. In addition, the researchers approached specific PCs with experience in KS directly using purposive sampling based on profession, setting (mental health care and long-term care) and years of experience in the field of KS.

The focus group meeting was moderated by M.D., observed by J.R. and video-recorded in order to enable the production of the verbatim transcript of this group discussion. A short background of the research was provided, and a topic list was used that covered signals of apathy, attitudes towards apathy, interaction with people with KS and apathy, and strategies to manage apathy. Furthermore, the main findings from the interviews were presented to deepen understanding from the perspective of the PC. Finaly, the original SABA program was presented to gather recommendations for the adjustment of the SABA program.

### 2.4. Analysis

The audio and video recordings were transcribed verbatim using the human-made transcription service provided by Amberscript (www.amberscript.com) and pseudonymized by J.R. The transcripts were stored and analyzed using Atlas.ti (version 9.18.00). Following the methods for thematic analysis described by Braun and Clarke [18], two researchers (M.D. and J.R.) independently derived codes from the data using open coding and discussed them until consensus was reached. Then, both researchers independently grouped and arranged the codes in themes and subthemes and discussed agreements and disagreements and the hierarchy until consensus was reached. In this phase, the perspective of the different participant groups (PwKS, FC, PC) were analyzed separately. Throughout the analysis process, discussions between the researchers were logged, and thematic maps were made to document the identified themes and the connections between them. Then, the identified themes and subthemes were discussed with a third researcher (Y.R.). In this phase, we discussed the overlap between the different perspectives and merged the overlapping themes and subthemes. Furthermore, we discussed themes and subthemes for coherence and resorted and refined codes in an inductive way, until consensus was reached. Notably, there was substantial overlap between the perspectives of FC and PC. Therefore, we chose not to report each perspective separately. Despite differences in background, both FC and PC largely described similar experiences. As a result, we opted to interpret all findings collectively, allowing for a more integrated understanding of the data. Subsequently, the identified themes and subthemes were discussed with all co-authors to determine the extent to which they align with the themes and subthemes from the original SABA study [14], adopting a deductive approach. The theme and subtheme headings were formulated based on group consensus.

## 3. Results

### 3.1. Participants’ Characteristics

Six PwKS and apathy have been invited to participate in an interview. Three of them gave informed consent. Reasons for not participating were terminal illness (*n* = 1), comorbid depressive disorder (*n* = 1), and not being interested (*n* = 1). In addition to the input from the PwKS, a patient representative (PR; a member of the patient support organization) was invited. Furthermore, FCs and PCs involved in the care for the included PwKS were also invited. The PR, FCs and PCs who were invited to participate in an interview all gave written informed consent. During the period of May 2024 to November 2024, eight semi-structured interviews were conducted with PwKS and apathy (*n* = 3), their FCs (one sibling and one partner; *n* = 2) and involved PCs (*n* = 2) and one PR. More details on the participants in the interviews are presented in Table 1.

Twelve PCs in the field of KS gave written informed consent and subsequently participated in the focus group. The focus group meeting took place in June 2024. More details on the participants in the focus group are presented in Table 2.

### 3.2. Themes and Subthemes

Based on the semi-structured interviews and the focus group meeting, two themes were identified that appear to play an important role in the recognition, identification and management of apathy in individuals with KS, and give direction to the adjustments of the SABA program (Table 3). The themes and subthemes are discussed in detail below and illustrated by meaningful quotes.

#### 3.2.1. The Challenge to Appraise Signals

This theme included the challenge to correctly label the behavior that was observed as apathy, the notion that apathetic behavior is considered as ‘normal’ (e.g., as age-appropriate or due to somatic comorbidity), and the fact that apathy may present differently across different PwKS, complicating the identification.

*Apathy is not correctly labeled.* All PwKS, the PR, and the FCs reported being unfamiliar with the term apathy. However, they nevertheless described characteristic apathetic behaviors, which they found notable and relevant. The PR, for example, described apathy as ‘lethargy’ and ‘becoming isolated’. The participating PCs gave descriptions such as inactive behavior, the inability to show emotions, and the absence of social interaction.


*“Someone is present, yet absent. They’re physically part of the group but blankly stare ahead and don’t take action.” (Activity coordinator, man)*



*‘Sometimes when we’re in the living room, I think: you’re all sitting here together, but you don’t really see each other’ (Nurse, woman)*



*‘They do things on autopilot in a very neutral way, with little facial expression.’ (Nurse, man)*


Both participants who were familiar with the term apathy and those who were not, searched for reasons behind the apathetic behavior and mentioned explanations for the apathy. As a result, apathy was often mislabeled and misunderstood.


*‘She participates in several activities here. […] And she can remember them herself. But if she doesn’t feel like doing them, she suddenly pretends that she forgot them. (Partner of a participant with KS, man)*



*‘Someone who has maybe experienced some kind of trauma and as a result is no longer able to open up to their surroundings or connect with others.’ (Nurse, woman)*


*Normalization of apathy.* PwKS, the PR, FCs, and some PCs considered not being active all the time as normal behavior, that is, part of normal aging, the result of physical limitations, the lack of physical fitness or comorbid disorders such as Chronic Obstructive Pulmonary Disease (COPD), or due to the current circumstances. This contributed to an underestimation of apathy.


*‘At a certain point, you just know it and you don’t question this anymore, because you know that’s how it is so it becomes normal in your life. […] Doing less becomes normal.’ (PwKS, man)*


*‘I have a lot of pain. And I have a constant ringing in my ears. […] Some of the other people, yes, they sit like* [crosses arms, puts head down, closes eyes]. *And to me, it’s not like there’s a lot of conversation going on.’ (PwKS, woman)*

*Apathy has a specific presentation in PwKS.* Descriptions and illustrations of apathy as mentioned by the participants in the interviews and the focus group often concern the quiet and silent *PwKS* and apathy. Apathy is often described as ‘doing nothing’, ‘sitting still’, ‘not being mentally present’ or ‘not giving a response’. There was consensus among the participants that, in the KS population, restless thoughts and agitated behavior are also common in individuals who appear apathetic. These different presentations—a quieter presentation vs. a more loud and expressive presentation—might complicate the appraisal of signals of apathy.


*‘We also have people who sometimes become aggressive or agitated quite quickly. Or you have people who can’t really handle eating together.’ (Nurse specialist, man)*


#### 3.2.2. The Challenge to Assess the Needs of PwKS

This theme includes PwKS rejecting support, the absence of emotions that is inherent to the nature of apathy, the importance of a therapeutic relationship of trust, and the contrast between past lifestyle and present living conditions.

*PwKS often reject support.* Different PCs illustrated how a lack of illness insight in combination with confabulations—which are characteristic for KS—may lead to unusual conversations because PwKS often live in their own perceived reality. As a result, PwKS may find it difficult to understand why caregivers want to interfere with their lives. In daily practice, this often leads to a rejection of support by PwKS. The daily PC of a man with KS describes the recurring statements he makes to her:


*‘Who decides that I have to get out of bed right now? I am a grown man. […] I can decide for myself. […] The way you live your life*
*—don’t force that upon me and on the way I live my life.’ (Nurse, man, imitating a PwKS)*


One participant with KS described that it is difficult to assess someone’s needs during a conversation and that relying on what a person says, does or expresses can lead to a misinterpretation.


*‘Look, that doesn’t mean I do a lot, but I would like to, yes.’ (PwKS, man)*



*‘Usually, when someone says something, you have to assume the exact opposite, then you’ll get closest to the truth. […] It is very complicated. But it is a matter of intuition and it’s something everyone knows for themselves.’ Interviewer: ‘have you had any experience with this yourself? That you said ‘leave me alone’ but actually meant the opposite?’ ‘I think we’ve all experienced that, we all have.’ (PwKS, man)*


*The absence of emotional expression.* The PR, FCs, and PCs provided examples of how the absence of emotion expression increases the uncertainty about whether they are doing the right thing by activating PwKS. This emotional flattening may easily lead to the assumption that someone is not enjoying himself or is not willing to participate in the activity or interaction.


*‘He just doesn’t know what he wants. Basically, he doesn’t want anything. Then you have to leave him alone.’ (PR, man)*



*‘We had a cup of coffee there and he really enjoyed it -but you couldn’t see it in his facial expression. Afterwards and in the evening, I called him and asked him ‘Did you like it?’. He said he really loved it and had a great day. But he doesn’t show it.’ (Brother of a participant with KS, man)*


*The importance of a therapeutic relationship of trust.* PCs stated that it is helpful to have known someone over an extended period of time as PwKS, despite the profound memory disorder, then respond more positively to suggestions and are more likely to participate in the suggested activities.


*‘We often find that clients tend to be much more reserved or take less initiative in the beginning. But when you have become a familiar face, things seem to become a bit easier. They need a period to get used to you, to get to know you and build trust.’ (Specialist nurse, woman)*


*The contrast between past lifestyle and present living conditions.* PwKS, FCs, and the PR emphasized the impact of the onset of KS on the lives of PwKS. KS is not age-dependent and has a sudden onset. Moreover, the severe memory disorder and other cognitive and behavioral changes that characterize KS significantly interfere with daily functioning. Often, there is a sharp contrast between how PwKS used to live an independent life with a job, and how their lives changed after the onset of KS.

The PR explained that, in addition to the changes in daily life structures of PwKS, there are also notable changes in their future life perspectives. While PwKS may have experienced a sense of hopelessness or aimlessness in their lives, their current caregivers are mainly focused on formulating treatment goals and optimizing well-being and quality of life.


*‘When you have had no sense of future direction in your life for years, and you have found comfort in alcohol—that is something we see in PwKS, right? They still have a tendency to seek alcohol—and then fresh and energetic caregivers in their thirties, with a hands-on attitude becomes involved like: ‘Sir, you need to go to your day care activities’ then I would also think: ‘Go away!’.’ (PR, man)*


### 3.3. Adjustments to the SABA Program

The original SABA program consists of four phases (Identifying signals of apathy, Screening and diagnosing apathy, Managing apathy, and Evaluation/maintenance) and includes materials and procedures applicable throughout these phases [13]. Some materials provide practical information on apathy that can be applied in the care of a specific individual with apathy, while other materials are fillable and adaptable forms that support the recognition and management of apathy in a specific individual and facilitates collaboration among caregivers within a specific case. The phases and procedures of the original SABA program will be fully incorporated into the SABA-KS program. Based on a combination of the experiences of the co-authors—who were also involved in the development and implementation of the original SABA program (J.N., A.P., D.G., and R.T.C.M.K.)—and the results of the current study, specific adjustments in a selection of the materials are recommended.

Guided by the themes and subthemes that were retrieved from the data analysis, the materials were evaluated and the extent to which they aligned with the current results was determined. Where misalignments were identified, specific adjustments were found to be necessary. Figure 1 presents an overview of these recommended adjustments of the SABA program.

All participants noted that the original SABA program was designed for a population of people with dementia and recognized this in the materials. They expressed the need to make adjustments in the presented characters (PwKS are typically younger than people with dementia), life stage, and examples of activities. One of the female PwKS stated that she did not recognize herself in the main character of the original SABA program:


*‘Look, when I see this [image of an old lady with grey hair], I think: I don’t really feel the connection … I don’t dye my hair for nothing’ (PwKS, woman)*


One of the FCs stated that the stereotypical image of the female character with dementia does not align with the KS population:


*‘That’s just a stereotypical image… Man-woman, if you look at it that way, then the woman is demented, and there’s a sewing machine—an old-fashioned sewing machine—next to her. […] And when I look at the people here, at your institute, it’s a large group of men…’ (Partner of a participant with KS, man)*


Different participants stated that certain activities included in the *Activity List* of the SABA program are not suitable for the KS population, for example, because they are too passive or sensory-based (e.g., skipping a balloon together, washing hands or enjoying the smell of food when someone else is cooking). They mentioned that PwKS want to participate in activities that are meaningful to them and typical for normal daily living or that require mental effort.


*‘I’m doing jigsaw puzzles and I’m reading. I have a magazine, I’m reading French magazines. So I do try to keep my brain busy enough’ (PwKS, woman)*



*‘There’s also the daily things, and things that are a part of your life right? I mean, when you go home tonight- […] but you have to cook dinner, you have to do the dishes, maybe you have to do laundry etcetera. Those are important things. And he still does his own laundry. Those are definitely very important things to keep doing.’ (brother of a participant with KS)*



*‘But playing with a balloon or rolling a ball across the table, yeah—I think most of the clients would say: Are you kidding me?’ (Occupational therapist, woman)*


Furthermore, FCs and PCs described that it is challenging to answer questions about the past activities and interests of PwKS. In some cases, this is due to a lack of information—either because the residents themselves are unable to remember past activities, or because relatives were no longer involved in the lives of the PwKS. More frequently, participants stated that an inactive lifestyle was already present prior to the onset of the disorder, often because of the long-term alcohol addiction. However, even when past activities are known, resuming these activities is not always possible. Cognitive impairments may limit the ability of PwKS to engage in their past activities, or past activities may have been closely linked to the consumption of alcohol, making replacement of past activities necessary.


*‘I used to drink beer every afternoon, but I guess I will take a break from that for now. Look, it’s not like I was drinking all afternoon, but maybe I started with my first beer at 1 p.m. Even if you take it easy, from 1 p.m. to 1 a.m., that is still 12 h.’ (PwKS, man)*


Consequently, it is often necessary to explore new activities and interests for PwKS. This process requires a creative and exploratory approach, in which trial and error may be essential to determine suitable activities for an individual resident.


*‘I often ask myself, when looking at apathy—the type of clients that are literally just staring in front of them. When you look at their past lives, they were people who may have been staring into nothing for the last 10 years at home as well…’ (Specialist nurse, woman)*


A complete overview of the proposed materials of the SABA-KS program is presented in Figure 2. An illustration of the *Discussion Guide*, adjusted for PwKS, is presented in Appendix B. In this example, the main character was changed to a man, and his appearance was adjusted to look younger than the main female character of the original SABA program. In addition, based on the current results, it was deemed important to position the PwKS in the logo literally at the same level as his family and professional caregiver to acknowledge and respect the patient’s need for autonomy and independence. Finally, the illustrated activities across the different materials were adjusted into activities that are suitable and meaningful for a younger population and represent ordinary daily life (e.g., grocery shopping, cooking and gardening).

## 4. Discussion

The aim of this study is twofold: (1) explore what is important in identifying and managing apathy in PwKS, and (2) examine appropriate adjustments for the SABA program to create a KS tailored version of this program. Two main themes were identified that are important for the different stakeholders: (1) the challenge to appraise signals of apathy, and (2) the challenge to assess the needs of PwKS. The SABA program partially addressed the themes that appeared to be important. However, the original SABA program was specifically developed for individuals with dementia living in nursing homes. Consequently, the content of that program aligns with the final life stage, and the layout reflects the target population of older people with dementia. Although the design of the SABA program seems suitable for the KS population, adjustments are needed to tailor it for PwKS. Specific adjustments that were recommended include the illustration of a more present and expressive resident who nevertheless fails to act on his plans, the addition of illustrative examples of apathy in KS, the revision of the diagnostic criteria for apathy, attention to the exploration of new activities, the addition of meaningful (daily) activities, and the addition of specific recommendations for the approach of PwKS.

The current results that illustrate the need for a specific intervention aimed at apathy seem at first sight contradictory with the results of a recent qualitative study on apathy treatment in dementia care [19]. That study concludes, based on their qualitative analysis, that apathetic residents benefit from basic nursing care practices and a patient-centered approach that dementia care nurses are already familiar with. However, nurses caring for residents with dementia in that study explained that this basic care includes awareness of and attention to signals of apathy, knowledge of how to diagnose apathy, and a patient-centered approach to managing apathy. Furthermore, their results highlight the importance of collaboration between family and professional caregivers [19]. In summary, the elements identified by that study that align with the phases of the SABA program are assumed to be basic care, and their perspective on good care aligns with the findings of our study. However, the result of our study shows that these elements are not yet part of the usual care aimed towards apathetic PwKS.

One of the strengths of the current study is the inclusion of diverse participants, which facilitated the exploration of experiences from different perspectives, including PwKS themselves. The participation of PwKS in research is relatively rare in the field of KS, due to the characteristic lack of illness insight in combination with the severe cognitive deficits and neuropsychiatric symptoms. Furthermore, the focus group consisted of KS professionals with varied backgrounds and experience. Recruitment from different types of organizations and purposive sampling contributed to this strength. Furthermore, the data triangulation in this study, with different types of qualitative methods (e.g., semi-structured interviews and a focus group), further strengthened its quality. Another strength is that the analyses have been performed in duplicate to optimize the validity of the results.

Although the PwKS formed an important part of our sample, the information that could be obtained from these interviews was limited. This was inherent to the apathy itself (two of the PwKS used very short sentences in their speech or sometimes did not answer the questions at all), related to a lack of disease awareness that is characteristic for KS, or due to the comorbid health problems. For instance, the PwKS shared assumptions about experiences or interpretations of the behavior of other residents, but could not were unable to reflect on their own behavior. To overcome this as much as possible, we purposely also interviewed FCs and PCs who were directly involved in the care for the included PwKS. Furthermore, it should be noted that the number of people who participated in an interview was limited. In qualitative research, however, the aim is not to obtain statistical generalizability, but rather to achieve depth, richness of data, and conceptual saturation [20]. The information gathered from our participants overlapped to such an extent that we were confident we had a clear enough picture to proceed to the next research phase. Ideally, we would have returned to the interview groups (PwKS, FC, and PC) after conducting the focus group to further explore and confirm the themes. However, the inclusion of the focus group did allow for a thorough exploration of the themes identified in the earlier interviews. Most relevant themes appear to have been captured, but we acknowledge that we cannot definitively confirm saturation.

Even though the proposed SABA-KS program may be promising for PwKS, its feasibility must be established empirically in future research. Subsequently, a randomized controlled trial (RCT) or a single-case experimental design (SCED) should be performed to establish the program’s efficacy. Given the individualized nature of the SABA-KS program, the SCED may be preferred, because this design is helpful to determine the optimal treatment for a specific individual, and enables the examination of treatment effects on an individual level [21]. Lane-Brown and Tate [22] recently demonstrated the successful application of the SCED in a study on the treatment of apathy in an individual with traumatic brain injury, by setting specific goals and measuring effects over time using a multiple-baseline approach. Challenges for future research include identifying appropriate instruments to measure the effects of the intervention. Instruments to assess apathy, such as the Apathy Motivation Index (AMI) [23,24] or Apathy Evaluation Scale (AES) [25], are available and have already been applied in PwKS [26,27]. However, measures such as increased competence in FCs and PCs or a reduced caregiver burden may also be desirable as outcome indicators of an effective intervention on apathy.

## 5. Conclusions

This study aims to explore important factors in identifying and managing apathy in PwKS, and to subsequently suggest specific adjustments for the SABA program in order to create a version of this program tailored for PwKS, that is, the SABA-KS program. The following main conclusions can be drawn from the results of this study. First, PwKS do not recognize apathy in themselves and, consequently, do not see the need for a specific intervention. They do recognize apathy in other residents and suggest that an intervention might be helpful for others. Furthermore, PwKS do not identify themselves with people with dementia, and should activation be deemed necessary by others, they want to participate in activities that are meaningful to them. Second, the participating FCs expressed that they were not familiar with the concept of apathy. They also stated that they found it tough to deal with their inactive loved ones with KS. Finally, PCs emphasized that it is difficult to appraise signals of apathy in their clinical practice and that they face challenges in activating and motivating apathetic PwKS to become more active. These findings have resulted in specific recommendations for the SABA-KS program, a modified version of the SABA program [13] that appeared applicable and feasible in the dementia population. The feasibility and efficacy of the SABA-KS program need to be established in future research.

## Figures and Tables

**Figure 1 geriatrics-10-00146-f001:**
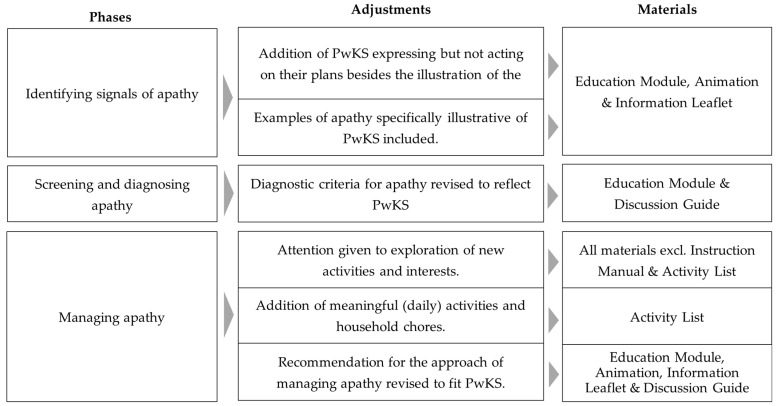
An overview of the necessary adjustments of the original SABA materials for developing the SABA-KS program and the specific materials for which the adjustments are proposed. Changes to illustrations or layouts are not specified. Note that the adjustments only refer to the first three phases of the program, since no adjustments to the Evaluation/maintenance phase were deemed necessary. PwKS = People with Korsakoff’s syndrome.

**Figure 2 geriatrics-10-00146-f002:**
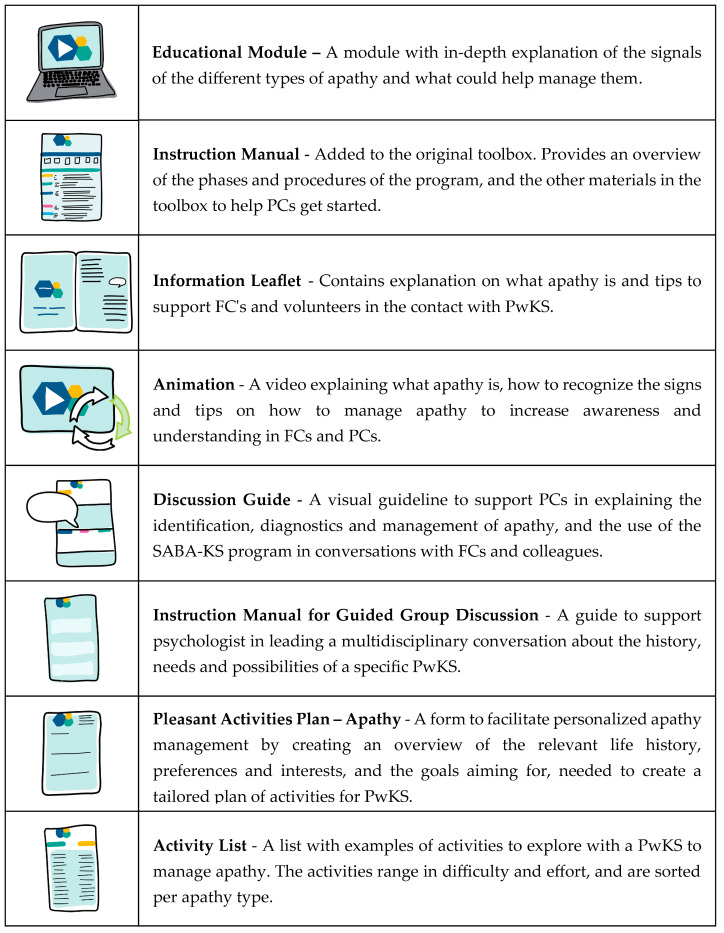
An overview of the materials of the Shared Action for Breaking through Apathy—Korsakoff’s syndrome (SABA-KS). PwKS = People with Korsakoff’s syndrome; FC = Family Caregiver; PC = Professional Caregiver.

**Table 1 geriatrics-10-00146-t001:** Characteristics of the eight participants in the interviews.

		Type of Organization	Age	Sex	Educational Level *
1	Person with KS	Mental health care	65	M	Average
2	Person with KS	Long-term care	65	F	High
3	Person with KS	Long-term care	77	M	Average
4	Patient representative	Patient support organization	70	M	High
5	Family caregiver (brother)	Mental health care	59	M	High
6	Family caregiver (partner)	Long-term care	72	M	High
7	Professional caregiver	Mental health care	54	F	Average
8	Professional caregiver	Long-term care	32	M	Average

* Educational level assessed using a Dutch classification range from 1 (less than primary school) to 7 (university degree). Educational level 1–3 is labeled as ‘low’, 4 and 5 are labeled as ‘average’, and 6 and 7 are labeled as ‘high’.

**Table 2 geriatrics-10-00146-t002:** Characteristics of the 12 professional caregivers included in the focus group.

	*n* = 12
Sex	
Man	6
Woman	6
Profession	
Activity coordinator	2
Specialist nurse	2
Care manager	1
Healthcare psychologist	2
Occupational therapist	1
Physiotherapist	1
Psychomotor therapist	1
Social worker	1
Spiritual counselor	1
Type of institution employed	
Mental healthcare organization	7
Long-term care organization	5
Years of experience in the field of KS	
<10 years	5
10–20 years	3
>20 years	4

**Table 3 geriatrics-10-00146-t003:** Overview of themes and subthemes.

Themes	Subthemes
The challenge to appraise signals	Apathy is not correctly labeled Normalization of apathy Apathy has a specific presentation among PwKS
The challenge to assess the needs of PwKS	PwKS often reject support The absence of emotional expression The importance of a therapeutic relationship of trust The contrast between past lifestyle and present living conditions

## Data Availability

The anonymized transcripts and Atlas.ti data files are available from the corresponding author upon reasonable request, as the participants did not give permission to share this data in a public repository.

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
