# Peer review of "The Development of a Program to Identify and Manage Apathy in Residents with Korsakoff’s Syndrome: A Qualitative Exploration of Patient, Family Caregiver, and Professional Caregiver Perspectives"

_geriatrics, 2025, doi:10.3390/geriatrics10060146_

Round 1
Reviewer 1 Report
Comments and Suggestions for Authors
Thanks for the opportunity to review your paper. This is an interesting study which revealed what is important in identifying and managing apathy in people with Korsakoff’s syndrome.
Comments for consideration:
Abstract - The results could be more detailed with inclusion of things like how many people participated and characteristics.
Methods - There should be more details about the researchers characteristics and reflexivity. How many interviews were conducted face-to-face versus video call? Could there be a difference in the information extracted since there may be less evident non-verbal cues in video call?
Discussion - Are there other limitations? For example, did the 8 participants really offer sufficient perspectives? Was there any difference in information obtained from face-to-face versus video call? Could the researchers experience etc. have influenced the findings?
Author Response
Comment 1. Abstract - The results could be more detailed with inclusion of things like how many people participated and characteristics.
Response 1. We appreciate the reviewer’s comment and see our oversight in omitting these details. We have added the following to the revised Abstract:
“Methods: This qualitative study consisted of semi-structured interviews with people with KS (n=3), family caregivers (one spouse and one sibling) and professional caregivers (two nurses), and a multidisciplinary focus group meeting with professional caregivers (n=12) experienced in care for people with KS.” Lines 28-31
Comment 2. Methods - There should be more details about the researchers' characteristics and reflexivity. How many interviews were conducted face-to-face versus video call? Could there be a difference in the information extracted since there may be less evident non-verbal cues in video call?
Response 2. We thank the reviewer for calling attention to the background of the respective researchers and their possible biases, which are indeed important factors in qualitative research. We have added a section dedicated to these points to the revised method section:
“Two interviews (with a PwKS and their patient representative) were led by M.D., a trained interviewer and psychologist with seven years of experience working with PwKS, and supported by J.R., a trained qualitative researcher with no prior experience with PwKS. The other six interviews were conducted by J.R. alone to limit interrater variability. M.D.’s clinical knowledge and understanding of PwKS were combined with J.R.’s research perspective, creating discussions leading to a balance between interpretations and deductions, and objective findings from the conversational data. Neither M.D. nor J.R. had a therapeutic relationship with the participants.” (lines 150-157).
Also, we recognize the point the reviewer has made regarding the interview settings and that it was only specified that the PwKS interviews were all done face-to-face. First, we would like to note that scheduling interviews with family caregivers and the patient representative—who were located across the country—as well as with professional caregivers with flexible working hours, was at times challenging. Although we have tried to conduct the interviews in the most optional way, we recognize that biases may be present. Regarding the possible biases of the settings themselves, all interviews were performed by or conducted in the presence of researcher J.R. to reduce inter-rater variability as much as possible. Notes of any relevant observations were taken during each interview. Furthermore, both interview modalities were carried out in a formal, non-casual setting. Where the video calls lacked the real-life factor, the face-to-face interviews were done in conference rooms at the respective care facilities. The details of the interview settings have been added to the manuscript:
“The interviews with all PwKS and one PC were conducted face-to-face. The other PC, the PR and both FCs were interviewed over video call. For the video calls the par-ticipants were asked to find a quiet and private area in their respective homes or workplaces. The interviewers conducted the calls from their office and had a neutral background. In all interviews, the researchers explicitly created a setting in which par-ticipants were stimulated to share all their relevant experiences and thoughts. Notes of any relevant observations were taken during each interview” (lines 143-149).
Comment 3. Discussion - Are there other limitations? For example, did the 8 participants really offer sufficient perspectives? Was there any difference in information obtained from face-to-face versus video call? Could the researchers experience etc. have influenced the findings?
Response 3. We thank the reviewer for addressing the importance of disclosing possible study limitations. We used purposive sampling to ensure a diverse study sample. That is, PwKS from different genders, ages and educational backgrounds were included from different care settings (from a mental health care clinic and from a nursing home), FC with different relationships to the residents, and PCs with different genders, ages, roles, occupations and amount of experience in the KS field. Furthermore, PwKS often present with somatic and psychiatric comorbidity, and commonly have strained relationships with loved ones. This meant that the number of residents who met the inclusion criteria was limited. For example, only 3 out of the 6 PwKS who were invited to participate in an interview gave informed consent. Also, few residents had loved ones who were still involved, and willing or able to be interviewed. While true saturation can never be guaranteed, we argue that the information gathered from our participants overlapped to such an extent that we were confident we had a clear enough picture to proceed to the next phase of the research. We have added this statement to the revised discussion:
“Furthermore, it should be noted that the number of people who participated in an interview was limited. In qualitative research, however, the aim is not to obtain statistical generalizability, but rather to achieve depth, richness of data, and conceptual saturation [22]. However, the information gathered from our participants overlapped to such an extent that we were confident we had a clear enough picture to proceed to the next research phase. Ideally, we would have returned to the interview groups (PwKS, FC, and PC) after conducting the focus group to further explore and confirm the themes. However, the inclusion of the focus group did allow for a thorough exploration of the themes identified in the earlier interviews. Most relevant themes appear to have been captured, but we acknowledge that we cannot definitively confirm saturation” (lines 489-498).
Reviewer 2 Report
Comments and Suggestions for Authors
Please avoid repetitions of a section in other section e.g. materials and methods in the abstract and introduction.
The title is not consistent with the paper’s content as in the later more focused on the FC and PC than the PwKS, it is seriously lacking objectivity as the study was not based on the patients’ signs and symptoms yet more to what have been faces by the FC and PC subjectively. The very small numbers of patients, n=3, and the FC plus PC (n=5, and 12) would heighten the power of study’s result, especially in the qualitative study design. The items in SABA for FC and PC and the recommended adjustments have not calculating the language for FC as lay persons, who are speculatively, not having a professional health and/ or medical background to even understand the pathophysiology of KS, furthermore apathy in KS is a complex neuropsychiatry disorder related with cognitive and memory functions. The biological alteration of the brain, i.e. hippocampal formation, limbic system, basal ganglia and the prefrontal cortex in the PwKS are not explored at all whilst this should give a more thorough understanding for the PC on the apathy underlying mechanism in each case. There are no data on the comorbidity of each case, the report of specific formal dan trainings of each FC and PC also the reasons why the authors didn’t controlled the gender, position/role and institution types of these caregivers, as these potentially increased biases. The recording voices could be improved using video instead, in this time of digital technology, voice only recordings are questionable choice, especially when deals with apathy. The PwKS and the caregivers responses should be recorded visually, by protecting the anonymity, respectively. The methods, results and discussion need a major improvement to answer, not only your first research question, but also your second one, as the later is not covered adequately and has yet reached scientific soundness. Although it has been stated to follow the SRQR list, the validity and reliability tests of each questionnaire item and the adjusted version and efforts to improve the SABA method had not been justified enough. The missing control group, i.e. from other cases of apathy not due to KS., adds serious bias. The conclusion, the first sentence, didn’t represent the main study aims and results; instead of stating about the PwKS perception of apathy (which supposedly was taken from only 3 chronic PwKS with apathy) rather than from the caregivers’ as shown in the whole full text. The FC must be analysed separately from the PC, as they have totally different background, education, role and experience when deal with PwKS (and these issues are not explained in the paper also).
Author Response
Comment 1. The title is not consistent with the paper’s content as in the later more focused on the FC and PC than the PwKS, it is seriously lacking objectivity as the study was not based on the patients’ signs and symptoms yet more to what have been faces by the FC and PC subjectively. The very small numbers of patients, n=3, and the FC plus PC (n=5, and 12) would heighten the power of study’s result, especially in the qualitative study design.
Response 1. We appreciate the reviewer’s comment that most information was provided by family and professional caregivers, and that the title might not fully reflect this. We adjusted the title of the paper to: The Development of a Program to Identify and Manage Apathy in Residents with Korsakoff’s Syndrome: A Qualitative Exploration of Patient, Family Caregiver, and Professional Caregiver Perspectives.
However, we do not want to exclude the PwKS perspectives from the title. Although the sample of individuals with Korsakoff’s syndrome (PwKS) included in this study is small, we deliberately aimed to include their perspectives, which are often not taken into account. We consider it a meaningful and innovative aspect of this study that, despite it being limited, we have included the views of PwKS themselves. Including the perspective of PwKS in research is unique, because limited or absent illness insight is a core characteristic of KS. Furthermore, including PwKS in research is challenging because of the severe cognitive impairments and comorbid somatic factors and neuropsychiatric symptoms. For example, of the six potential participants in our study, three were unable to take part due to terminal illness, severe depression, or a lack of interest. Regarding this comment, we have emphasized the added value of including the perspective of PwKS in our revised discussion:
“One of the strengths of the current study is the inclusion of diverse participants, which facilitated the exploration of experiences from different perspectives, including PwKS themselves. The participation of PwKS in research is relatively rare in the field of KS, due to the characteristic lack of illness insight in combination with the severe cognitive deficits and neuropsychiatric symptoms. Furthermore, the focus group consisted of KS professionals with varied backgrounds and experience. Recruitment from different types of organizations and purposive sampling contributed to this strength.” (lines 470-476).
Comment 2.1. The items in SABA for FC and PC and the recommended adjustments have not calculating the language for FC as lay persons, who are speculatively, not having a professional health and/ or medical background to even understand the pathophysiology of KS, 2.2. Furthermore apathy in KS is a complex neuropsychiatry disorder related with cognitive and memory functions. The biological alteration of the brain, i.e. hippocampal formation, limbic system, basal ganglia and the prefrontal cortex in the PwKS are not explored at all whilst this should give a more thorough understanding for the PC on the apathy underlying mechanism in each case.
Response 2.1. We thank the reviewer for emphasizing that providing clear and understandable information on KS and apathy should be an essential part of SABA. In fact, providing such information, also to lay persons, is already an important part of both the original SABA program and KS-tailored version of the SABA program. Regarding SABA-KS, an informative animation has been designed for professional caregivers and family members who are not overly familiar with apathy in PwKS. This animation provides background information on apathy as a neuropsychiatric symptom of KS and illustrates different ways in which apathy can present and be recognized in different individuals with KS. Furthermore, practical tips for the interaction with PwKS and apathy are provided and illustrated. In addition, the SABA program also includes an information leaflet designed for family members who are not familiar with KS and apathy. This information leaflet provides basic and simply formulated information on apathy in PwKS, as well as tips for the interaction with PwKS. Finally, different materials of the SABA program (e.g. the Discussion Guide and the Pleasant Activities Plan – Apathy) facilitate conversations between professional caregivers and family members on recognizing and managing apathy in a specific individual with KS. We now realize that this was not clarified enough in the initial manuscript. We have added the following text to the revised introduction:
“The SABA program is a theory- and practice-based intervention to empower family (FC) and professional (PC) caregivers to identify and manage apathy in individuals with dementia living in long-term care facilities. The program provides understandable information and enhances awareness of apathy, helps to overcome a lack of knowledge - including inadequate expectations, and supports the skills of caregivers and family members to manage apathy.” (lines 80-85).
Regarding the second point of this comment, it must be stressed that the SABA program is a behavioral intervention and that the professional caregivers that work with this program mainly consists of nurses with a practical educational background. Although enhancing knowledge is one of the targets of the SABA program, it is important that the knowledge provided is practical, making sure it can be implemented in their daily work with PwKS, therefore we did not include detailed information on biological alterations of the brain.
Comment 3. There are no data on the comorbidity of each case, the report of specific formal and trainings of each FC and PC also the reasons why the authors didn’t controlled the gender, position/role and institution types of these caregivers, as these potentially increased biases.
Response 3. In contrast to quantitative studies, qualitative studies cannot be controlled in a comparable way. In order to compensate for this, we intentionally searched for participants that differed in age, sex, position/role, educational level, years of working experience in the field of KS (PCs) and the type of institution they are living in (PwKS) or are employed at (PCs). In this way, we tried to present as broad a picture as possible.
We apologize that the we did not provide much information on the participants. We have added the following to the manuscript:
“PwKS and apathy were recruited from two clinical institutes specialized in KS care, located in the southern part of the Netherlands: one being a diagnostic mental healthcare clinic and long-term care unit, the other a center of expertise of a long-term care organization. Primary practitioners of the different units were asked to select potential participants. One inclusion criterium was a diagnosis of alcoholic KS. That is, patients had to meet the DSM-5 criteria for Alcohol-induced Major Neurocognitive Disorder, amnesic-confabulatory type (291.1; i.e., Korsakoff’s syndrome [18]), verified by neuropsychological assessment. In addition, the criteria for alcoholic KS [19]) had to be met; that is, all were alcoholic KS patients who had a history of malnutrition or thiamine deficit (with suspicions of a Wernicke encephalopathy), and a disproportionate memory disorder compared to all other cognitive domains. The impairments were not attributable to another medical condition or use of other substances. Participants had a verified history of excessive, chronic alcohol use. Also, patients had to show signs of apathy observed by an FC and/or PC. Exclusion criteria for this study were the presence of active psychiatric disorders that might interfere with cognitive functioning (e.g. major depressive disorder, psychotic spectrum disorder). All patients were at least 6 weeks abstinent from alcohol. Participants were recruited using purposive sampling based on sex, setting (mental health care and long-term care) and educational level. After including an individual with KS and apathy, one FC and one PC of that patient were also asked to participate. Formal caregivers were a nurse practitioner and a nursing assistant.” (Lines 123-142)
Comment 4. The recording voices could be improved using video instead, in this time of digital technology, voice only recordings are questionable choice, especially when deals with apathy. The PwKS and the caregivers responses should be recorded visually, by protecting the anonymity, respectively.
Response 4. All interviews were conducted eighter face-to-face or by video call to enable the researcher to observe the interviewee. Notes of any relevant observations were taken during each interview and both interview modalities were carried out in a formal, non-casual setting.
As stated, the face-to-face conducted interviews were additionally audio-recorded to enable the production of verbatim transcripts, which formed the basis for deriving relevant codes and subsequently identifying key themes and subthemes. Audio recordings were deemed sufficient to meet these ends. As recommended by the Institutional Review Board, we had to be careful not to collect unnecessary personal data. Therefore, we recorded the interviews by video only if face-to-face conduction was not possible.
The focus group was also conducted face-to-face, and this meeting was video-recorded in order to keep an overview of who responded to whom and to distillate different reactions in case participants spoke simultaneously. We agree that this was not included explicitly in the manuscript. We have added the following to the revised methods: “Notes of any relevant observation were taken during each interview.” (lines 148-149), and “The focus group meeting was moderated by M.D., observed by J.R. and video-recorded in order to enable the production of the verbatim transcript of this group discussion.” (lines 170-172), and “The audio and video recordings were transcribed verbatim using the human-made transcription service provided by Amberscript (www.amberscript.com) and pseudonymized by J.R.” (lines 179-181).
Comment 5. The methods, results and discussion need a major improvement to answer, not only your first research question, but also your second one, as the latter is not covered adequately and has yet reached scientific soundness. Although it has been stated to follow the SRQR list, the validity and reliability tests of each questionnaire item and the adjusted version and efforts to improve the SABA method had not been justified enough. The missing control group, i.e. from other cases of apathy not due to KS., adds serious bias.
Response 5. We thank the reviewer for addressing this point. Based on this comment, we realize that the form and content of the SABA-KS program were not explained explicitly enough. Neither of the materials of the SABA-program (presented in Figure 2) are questionnaires. The materials with entry fields, such as the Pleasant Activities Plan – Apathy or the Instruction Manual for Guided Group Discussion, should be considered adaptable forms that, for example, can be used to facilitate the conversation between caregivers on recognizing signals of apathy in a specific resident or that supports working together by establishing working agreements and optional activities for a specific resident. The following adjustments have been made to the revised results:
“The original SABA program consists of four phases (Identifying signals of apathy, Screening and diagnosing apathy, Managing apathy, and Evaluation/maintenance) and includes materials and procedures applicable throughout these phases [15]. Some materials provide practical information on apathy that can be applied in the care of a specific individual with apathy, while other materials are fillable and adaptable forms that support the recognition and management of apathy in a specific individual and facilitates collaboration among caregivers within a specific case. The phases and procedures of the original SABA program will be fully incorporated into the SABA-KS program. Based on a combination of the experiences of the co-authors – who were also involved in the development and implementation of the original SABA program (J.N., A.P., D.G. and R.T.C.M.K) – and the results of the current study, specific adjustments in a selection of the materials are recommended.” (lines 347-358).
Furthermore, it should be noted that this is an article on the development of a novel intervention. As we state in our discussion, the feasibility and subsequently the efficacy of the SABA-KS program should be established in future research. We surely do not want to imply that conclusions on the feasibility or the efficacy of the SABA-KS program can be made based on this study.
Comment 6. The conclusion, the first sentence, didn’t represent the main study aims and results; instead of stating about the PwKS perception of apathy (which supposedly was taken from only 3 chronic PwKS with apathy) rather than from the caregivers’ as shown in the whole full text.
Response 6. We thank the reviewer for this comment and have made the following adjustments to the revised Conclusions in order to optimize the quality and readability:
“This study aims to explore important factors in identifying and managing apathy in PwKS, and to subsequently suggest specific adjustments for the SABA program in order to create a version of this program tailored for PwKS, that is, the SABA-KS program. The following main conclusions can be drawn from the results of this study. First, PwKS do not recognize apathy in themselves and, consequently, do not see the need for a specific intervention. They do recognize apathy in other residents and suggest that an intervention might be helpful for others. Furthermore, PwKS do not identify themselves with people with dementia, and should activation be deemed necessary by others, they want to participate in activities that are meaningful to them. Second, the participating FCs expressed that they were not familiar with the concept of apathy. They also stated that they found it tough to deal with their inactive loved ones with KS. Finally, PCs emphasized that it is difficult to appraise signals of apathy in their clinical practice and that they face challenges in activating and motivating apathetic PwKS to become more active. These findings have resulted in specific recommendations for the SABA-KS program, a modified version of the SABA program [15] that appeared applicable and feasible in the dementia population. The feasibility and efficacy of the SABA-KS pro-gram need to be established in future research.” (lines 515-531).
Comment 7. The FC must be analysed separately from the PC, as they have totally different background, education, role and experience when deal with PwKS (and these issues are not explained in the paper also).
Response 7. We fully agree that it is important to look at the different perspectives separately, as they might yield different results. We did analyze the data obtained from the different stakeholders separately, but agree with the reviewer that this is not stated clearly enough in our manuscripts yet. We have added the following to the revised Methods:
“Two researchers (M.D. and J.R.) independently derived codes from the data using open coding and discussed them until consensus was reached. Then, both researchers independently grouped and arranged the codes in themes and subthemes and discussed agreements and disagreements and the hierarchy until consensus was reached. In this phase, the perspective of the different participant groups (PwKS, FC, PC) were analyzed separately. Throughout the analysis process, discussions between the researchers were logged, and thematic maps were made to document the identified themes and the connections between them. The identified themes and subthemes were discussed with a third researcher (Y.R.). In this phase, we discussed the overlap between the different perspectives and merged the overlapping themes and subthemes. Furthermore, we discussed themes and subthemes for coherence and resorted and refined codes in an inductive way, until consensus was reached. Notably, there was substantial overlap between the perspectives of FC and PC. Therefore, we chose not to report each perspective separately. Despite differences in background, both FC and PC largely described similar experiences. As a result, we opted to interpret all findings collectively, allowing for a more integrated understanding of the data.” (lines 184-199).
Reviewer 3 Report
Comments and Suggestions for Authors
Thank you very much for the opportunity to review this manuscript. I have read it carefully and appreciate the authors’ efforts in addressing an important and underexplored topic. However, I believe the manuscript presents several significant challenges that limit its suitability for publication at this stage.
1)This manuscript explores the applicability of the Shared Action for Breaking through Apathy (SABA) program—originally developed for individuals with dementia—to nursing-home residents with Korsakoff’s syndrome (KS). While the topic is relevant and potentially valuable given the prevalence of apathy in KS populations, the study presents several critical limitations that hinder its suitability for publication in its current form.
2)The manuscript’s central aims—to explore how apathy is identified and managed in KS, and to investigate necessary adaptations to the SABA program—are not clearly delineated throughout the paper. The rationale for selecting the SABA program as a foundation is not sufficiently justified, nor is the theoretical framework that underpins its adaptation for KS. As a result, the reader is left uncertain about the conceptual bridge between dementia-related apathy and KS-specific manifestations, which may differ significantly in etiology and behavioral expression.
3)The study employs a qualitative design using semi-structured interviews and a single focus group. While qualitative methods are appropriate for exploratory research, the sample size is notably small and lacks diversity. There is limited information about participant selection criteria, demographic characteristics, and the saturation of themes. Furthermore, the thematic analysis, though mentioned, is not described in sufficient detail to assess its rigor. The coding process, inter-rater reliability, and validation strategies are absent, raising concerns about the credibility and reproducibility of the findings.
4)The results section identifies two broad themes: (1) the challenge of appraising signals of apathy, and (2) the difficulty in assessing the needs of individuals with KS. However, these themes are presented in a generalized manner, without rich illustrative data or nuanced interpretation. The discussion does not critically engage with existing literature on KS or apathy, nor does it offer a compelling argument for how the findings advance current understanding or practice. The proposed adjustments to the SABA program are vague and lack operational detail, making it difficult to evaluate their feasibility or relevance.
5)The study concludes by suggesting that the adapted SABA program may be feasible for KS populations, but this assertion remains speculative. No pilot implementation or outcome data are provided. Without empirical validation, the practical utility of the proposed adjustments is uncertain. The manuscript would benefit from a clearer roadmap for future research, including specific hypotheses, measurable outcomes, and implementation strategies.
6)The manuscript suffers from structural inconsistencies and linguistic imprecision. Key concepts are introduced without adequate explanation, and transitions between sections are abrupt. The abstract, while informative, does not fully capture the limitations or scope of the study. Overall, the writing lacks the clarity and coherence expected of a publishable academic article.
7)The topic is promising, and the authors may consider conducting a more robust, mixed-methods study with a clearer theoretical foundation and empirical validation of the adapted intervention. Strengthening the methodological transparency and deepening the analysis would significantly enhance the manuscript’s contribution to the field.
Comments on the Quality of English Language
NA
Author Response
Comment 1. This manuscript explores the applicability of the Shared Action for Breaking through Apathy (SABA) program—originally developed for individuals with dementia—to nursing-home residents with Korsakoff’s syndrome (KS). While the topic is relevant and potentially valuable given the prevalence of apathy in KS populations, the study presents several critical limitations that hinder its suitability for publication in its current form.
The manuscript’s central aims—to explore how apathy is identified and managed in KS, and to investigate necessary adaptations to the SABA program—are not clearly delineated throughout the paper. The rationale for selecting the SABA program as a foundation is not sufficiently justified, nor is the theoretical framework that underpins its adaptation for KS. As a result, the reader is left uncertain about the conceptual bridge between dementia-related apathy and KS-specific manifestations, which may differ significantly in etiology and behavioral expression.
Response 1. We thank the reviewer for addressing this important point. Apathy is increasingly considered a transdiagnostic symptom. Although the clinical conditions in which apathy occur vary, brain abnormalities underlying apathy and the clinical presentation of apathy among different populations show substantial overlap. We have added the following to the revised Introduction to strengthen the rationale for the use of knowledge on apathy from the dementia population and for the selection of the SABA program for this study:
“There is broad consensus that apathy is a transdiagnostic symptom and that its clinical presentation is similar across different diseases or disorders [11]. A transdiagnostic view on apathy is beneficial for the improvement of fundamental research on the construct of apathy and for deepening the understanding of the underlying neuropathology.” (lines 61-64).
And
“In accordance with the recommendations formulated in our recent review [12], patient compliance with the SABA program does not rely on the intrinsic motivation of apathetic individuals themselves, as the program is directed at FCs and PCs who motivate and activate the residents when the apathy is present. Furthermore, the SABA program is designed as a customizable intervention that is shaped by personal features and preferences of the individuals with apathy. Finally, the recognition and assessment of apathy is a prominent part of the SABA program. We therefore expect that the SABA program, with specific adjustments, can also be applied in the care of PwKS who display apathy.” (lines 92-99)
Comment 2. The study employs a qualitative design using semi-structured interviews and a single focus group. While qualitative methods are appropriate for exploratory research, the sample size is notably small and lacks diversity. There is limited information about participant selection criteria, demographic characteristics, and the saturation of themes. Furthermore, the thematic analysis, though mentioned, is not described in sufficient detail to assess its rigor. The coding process, inter-rater reliability, and validation strategies are absent, raising concerns about the credibility and reproducibility of the findings.
We thank the reviewer for their comments regarding the validity and reliability of our study. In qualitative research, the aim is not to obtain statistical generalizability, but rather to achieve depth, richness of data, and conceptual saturation. We have recruited a sample of N=20 (see Tables 1 and 2). We considered this large enough to reach saturation, as our study aim was narrow, sample specificity was dense, the theoretical background was applied and, the dialogue was overall strong (Malterud et al., Sample size in qualitative interview studies: guided by information power. Qual Health Res 2015;26(13):1753-1760. doi:10.1177/1049732315617444). The participants in our study came from diverse backgrounds, which allowed for a variety of perspectives to be captured. We deliberately chose to first conduct individual interviews with PwKS, FCs, PCs, followed by a focus group. The purpose of the focus group was to confirm the themes identified in the earlier interviews, and to explore whether any new themes would emerge. This sequential design enabled a thorough exploration of the topic, and we argue that most relevant themes have been addressed. However, we acknowledge that we cannot confirm full thematic saturation.
We have added the following as a limitation to the revised manuscript:
“Furthermore, it should be noted that the number of people who participated in an interview was limited. In qualitative research, however, the aim is not to obtain statistical generalizability, but rather to achieve depth, richness of data, and conceptual saturation [22]. The information gathered from our participants overlapped to such an extent that we were confident we had a clear enough picture to proceed to the next research phase. Ideally, we would have returned to the interview groups (PwKS, FC, and PC) after conducting the focus group to further explore and confirm the themes. However, the inclusion of the focus group did allow for a thorough exploration of the themes identified in the earlier interviews. Most relevant themes appear to have been captured, but we acknowledge that we cannot definitively confirm saturation.” (lines p.489-498)
With regards to the participant selection criteria and demographic characteristics, we used the Standards for Reporting Qualitative Research (SRQR; O’Brien et al. Standards for reporting qualitative research: A synthesis of recommendations. Acad Med 2014, 89(9), 1245-1251. doi:10.1097/ACM.0000000000000388) and included this list with text references in Appendix A. These standards helped us to conduct and report our study carefully. We intentionally searched for participants who differ in age, sex, position/role, educational level, years of working experience in the field of KS (PCs), and the type of institution they are living (PwKS) or employed at (PCs). In this way, we aimed to present as broad a picture as possible. However, this comment shows that the information in the current manuscript is too limited yet. We have added multiple details throughout the revised Methods (lines 123-157).
Comment 3. The results section identifies two broad themes: (1) the challenge of appraising signals of apathy, and (2) the difficulty in assessing the needs of individuals with KS. However, these themes are presented in a generalized manner, without rich illustrative data or nuanced interpretation. The discussion does not critically engage with existing literature on KS or apathy, nor does it offer a compelling argument for how the findings advance current understanding or practice. The proposed adjustments to the SABA program are vague and lack operational detail, making it difficult to evaluate their feasibility or relevance.
Response 3. We agree with the reviewer that the embedding in existing literature is limited in the current manuscript. However, existing literature on apathy in KS is scarce and literature on effective treatment methods to reduce apathy in people with KS is lacking altogether. The current study builds upon earlier work from our research groups, and embedding in this line of research is clarified in the revised Introduction, as mentioned in our response to the first comment from Reviewer 3:
Regarding the last point in this comment, we would like to underline that this study is a first step towards a KS-tailored intervention. Although this study builds on recommendations identified in our recent literature review, it is important to mention that it is not the aim of this study to examine the feasibility or the efficacy of the SABA-KS program. As we state in our revised discussion, the feasibility and efficacy of the SABA-KS program must be established in future research. However, we recognize that the proposed adjustments to the original SABA program were not described in much detail in the initial manuscript and need further clarification. We have added the necessary information to Figure 1.
Comment 4. The study concludes by suggesting that the adapted SABA program may be feasible for KS populations, but this assertion remains speculative. No pilot implementation or outcome data are provided. Without empirical validation, the practical utility of the proposed adjustments is uncertain. The manuscript would benefit from a clearer roadmap for future research, including specific hypotheses, measurable outcomes, and implementation strategies.
Response 4. We fully agree with the reviewer that we cannot make a statement regarding the feasibility of the SABA-KS program. We have added more suggestions for future research to our revised Discussion:
“Even though the proposed SABA-KS program may be promising for PwKS, its feasibility must be established empirically in future research. Subsequently, a randomized controlled trial (RCT) or a single-case experimental design (SCED) should be performed to establish the program’s efficacy. Given the individualized nature of the SABA-KS program, the SCED may be preferred, because this design is helpful to determine the optimal treatment for a specific individual, and enables the examination of treatment effects on an individual level [23]. Lane-Brown and Tate [24] recently demonstrated the successful application of the SCED in a study on the treatment of apathy in an individual with traumatic brain injury, by setting specific goals and measuring effects over time using a multiple-baseline approach. Challenges for future research include identifying appropriate instruments to measure the effects of the intervention. Instruments to assess apathy, such as the Apathy Motivation Index (AMI) [25,26] or Apathy Evaluation Scale (AES) [27], are available and have already been applied in PwKS [28,29]. However, measures such as increased competence in FCs and PCs or a reduced caregiver burden may also be desirable as outcome indicators of an effective intervention on apathy.” (Lines 499-513)
And to the Conclusions:
“These findings have resulted in specific recommendations for the SABA-KS program, a modified version of the SABA program [13] that appeared applicable and feasible in the dementia population. The feasibility and efficacy of the SABA-KS program need to be established in future research.” (Lines 527-531)
Comment 5. The manuscript suffers from structural inconsistencies and linguistic imprecision. Key concepts are introduced without adequate explanation, and transitions between sections are abrupt. The abstract, while informative, does not fully capture the limitations or scope of the study. Overall, the writing lacks the clarity and coherence expected of a publishable academic article.
Response 5. We made multiple adjustments throughout the text in order to clarify key concepts and improve the readability of the article. For example, in our revised Introduction we clarified our view on apathy as a transdiagnostic symptom and argued in more detail why we selected the SABA program for the development of a KS-specific intervention by making the following adjustments:
“There is broad consensus that apathy is a transdiagnostic symptom and that its clinical presentation is similar across different diseases or disorders [11]. A transdiagnostic view on apathy is beneficial for the improvement of fundamental research on the construct of apathy and for deepening the understanding of the underlying neuropathology.” (lines 61-64)
And
“In accordance with the recommendations formulated in our recent review [14], patient compliance with the SABA program does not rely on the intrinsic motivation of apathetic individuals themselves, as the program is directed at FCs and PCs who motivate and activate the residents when the apathy is present. Furthermore, the SABA program is designed as a customizable intervention that is shaped by personal features and preferences of the individuals with apathy. Finaly, the recognition and assessment of apathy is a prominent part of the SABA program. We therefore expect that the SABA program, with specific adjustments, can also be applied in the care of PwKS who display apathy.” (lines 92-99)
Furthermore, we have expanded the limitations section and we made multiple other adjustments to the revised Discussion and the revised Conclusions in order to improve the coherence and readability of the text (see lines 489-513). Due to the 250-word limit for the abstract, we could not add more details on the limitations there, without having to remove other information that we deem essential.
Comment 6. The topic is promising, and the authors may consider conducting a more robust, mixed-methods study with a clearer theoretical foundation and empirical validation of the adapted intervention. Strengthening the methodological transparency and deepening the analysis would significantly enhance the manuscript’s contribution to the field.
Response 6. We thank the reviewer for this comment. We are obviously not able to change our method to a mixed-methods study design, since we have obtained ethical approval for the current study design and analysis plan. However, we argue that the revised version of our manuscript has improved the methodological transparency and has further clarified our analyses, thus offering a valuable contribution to the field.
Round 2
Reviewer 2 Report
Comments and Suggestions for Authors
The response of each comment is sound and well understood; to share the new approach on the quite complex case, hopefully this manuscript would also be well received.
Author Response
Comments and suggestions for authors: The response of each comment is sound and well understood; to share the new approach on the quite complex case, hopefully this manuscript would also be well received.
Response: We thank the reviewer for their positive response and appreciate their support. No further changes were needed.
Reviewer 3 Report
Comments and Suggestions for Authors
There are no more comments.
Author Response
Comments and suggestions for authors: There are no more comments.
Response: We thank the reviewer for their positive response.